# OpenReview forum: "Unitary Convolutions for Message-passing and Positional Encodings on Directed Graphs"
_ICML.cc/2026/Conference — ICML 2026 regular_

### Official Review · Reviewer_41ew · 2026-03-09

**Soundness:** 3
**Presentation:** 2
**Significance:** 3
**Originality:** 2
**Overall Recommendation:** 4
**Confidence:** 3

**Summary:**

This paper proposes a neural network for directed graphs with edge features via unitary convolutions, which prevent oversmoothing and feature collapse. To handle directed graphs, the paper proposes an Hermitian projection of the directed adjacency, while edge features are processed by a linear transformation that produces an edge weight. Theoretical results show that the proposed model is as expressive as the directed WL test and avoids oversmoothing, and experiments on multiple datasets confirm its effectiveness.

**Compliance With Llm Reviewing Policy:**

Affirmed.

**Final Justification:**

I thank the authors for their rebuttal and response, which have addressed most of my concerns. I believe the clarifications on (i) the connection with the CoED framework, (ii) the WL-test for edge-featured graphs and (iii) the contributions w.r.t. prior work improve the originality of the paper, so I have increased the originality score.

The additional results comparing methods with edge directionality, edge features or both clarify the interplay of these different sources of information, and justify the overall model. Considering these results, along with better descriptions of the experiments in the Appendix and in Table 3 and a more detailed explanation of the treatment of complex numbers and neural network implementation, I have increased my score to a weak accept. Since other results would be beneficial, but require more time than the rebuttal's deadline (e.g., the ablation results on datasets other than DSP and LUT, the additional edge feature-aware baselines PNA and ECC), and the paper requires several modifications to incorporate the clarifications provided by the authors, I do not believe this paper deserves a higher score.

**Key Questions For Authors:**

- How are the complex numbers in eq. (3) handled by the neural network?
- What is the impact of considering edge features on the expressivity and oversmoothing of Dune, and how can one characterize the interplay between edge directionality and features?
- How does the method perform against other edge feature-aware methods on directed graphs with edge features?

**Limitations:**

Yes

**Strengths And Weaknesses:**

**Strengths**

- The paper presents theoretical results to characterize the expressivity and resistance to oversmoothing of the proposed model.
- The experiments on directed graphs are comprehensive across datasets and baselines.

**Weaknesses**

- The contribution appears overall limited: the framework consists of the introduction of a directed adjacency matrix formulation and a mechanism to process edge features within the UniGCN framework proposed in [1]. However:
	- The inclusion of edge features in the model is not sufficiently discussed. Theoretical results only rely on the directed adjacency formulation, and the only experiments concerning edge features are in Appendix F.5. If the use of edge features is central in the proposed framework, the authors should address their impact on the theoretical analysis, analyze the interplay between edge directionality and edge features and provide more extensive experiments comparing against other edge feature-aware methods.
	- The theoretical results are immediate applications of existing results from [1] for oversmoothing and from [2] for expressivity. It would be beneficial if the authors could elaborate more on the additional insights introduced by their proposed formulation, especially compared to [1].

- The clarity of the presentation can be improved:
	- The parametrization in eq. (4) includes complex numbers, and it is not clear how these are handled in the neural network.
	- It is not specified how a complete neural network is defined in relation to the unitary convolution operation in eq. (5)
	- Notation inconsistency: the edge features are denoted with both $\mathbf{E}\_{ij}$ and $E_{ij}$


**References**

[1] Kiani, Bobak, Lukas Fesser, and Melanie Weber. "Unitary convolutions for learning on graphs and groups." _Advances in Neural Information Processing Systems_ 37 (2024): 136922-136961.

[2] Rossi, Emanuele, et al. "Edge directionality improves learning on heterophilic graphs." _Learning on graphs conference_. PMLR, 2024.

---

> ### Author Rebuttal · Authors · 2026-03-30
>
> We would like to thank the reviewer for their detailed and constructive feedback.
>
> > The inclusion of edge features in the model is not sufficiently discussed...
>
> Theory: Considering edge features does not affect (the absence of) oversmoothing in Dune, since we integrate edge features before mapping onto the unitary group, so the resulting adjacency matrix is both informed by edge features and unitary, hence (5) does not induce oversmoothing (Proposition (6)).
>
> The interplay between edge features, edge directionality, and expressivity is to the best of our knowledge not well understood. Our framework can be most cleanly connected to [1]. We can connect DUNE to their Continuous Edge Direction (CoED) regime by restricting the map $f_E$ in Eq. (7) so that for each edge pair $(i,j)$,
>
> $$(A_E)_{ij} = \mu_{ij} = \cos \theta_{ij}, (A_E)_{ji} = \mu_{ji} = \sin \theta_{ij}, \theta_{ij} \in [0,\pi/2],$$
>
> which is exactly the fuzzy-edge parameterization used in CoED, where
>
> $$\mu_{ji} = \sqrt{1 - \mu_{ij}^2}.$$
>
> In CoED, this coupling is what allows the real and imaginary parts of the fuzzy Laplacian to separately encode incoming and outgoing weighted messages, leading to expressivity up to a weak WL test for directed graphs with fuzzy edges.
> We will revise the paper to clarify that the edge-feature mechanism is partially grounded through this CoED connection. A full theoretical treatment of edge features, directionality, and expressivity is beyond this paper’s scope.
>
> [1] Pahng, Seong Ho, and Sahand Hormoz. "Improving Graph Neural Networks by Learning Continuous Edge Directions." The Thirteenth International Conference on Learning Representations.
>
> Experiments: please note that we already compare against common edge feature-aware GNNs (directed and undirected) in Tables 15 and 20. We agree with the reviewer that these could be more extensive and therefore provide the following additional results (d=edge direction, e=edge features, *edges removed from input graph).
>
> | Model |     DSP |     LUT |
> | ---------------------------------------- | ------: | ------: |
> | UniGCN | $2.609$ | $2.030$ |
> | DUNE (e) | $2.524$ | $1.917$ |
> | DUNE (d) | $2.316$ | $1.884$ |
> | DUNE (e, d) | $2.204$ | $1.753$ |
> | GIN | $3.405$ | $2.818$ |
> | GINE (e) | $3.068$ | $2.391$ |
> | Bi-GIN (d) | $2.637$ | $1.814$ |
> | Bi-GINE (e, d) | $2.482$ | $1.796$ |
> | GatedGCN* | $3.183$ | $2.696$ |
> | GatedGCN (e) | $2.831$ | $2.468$ |
> | Bi-GatedGCN (d) | $2.562$ | $2.085$ |
> | Bi-GatedGCN (e, d) | $2.417$ | $1.830$ |
>
> We compare against GatedGCN because it is a common edge-aware message-passing layer with a mechanism distinct from GINE. We report mean RMSE (lower is better).
>
> We will extend these comparisons to all datasets with edge features and edge directionality for the final manuscript.
>
> > The theoretical results...
>
> We only share the general proof structure of Proposition (3) with [2], which is a standard way of proving expressivity (see, e.g. the original GIN paper).
> For over-smoothing, we could not follow the proof of [1] as it uses special properties of the undirected setting (e.g. commutativity of operator with Dirichlet form matrix). The results also raise interesting differences with [1] (also see Table 1):
> - Expressivity, especially for directed graphs ([1] does not discuss expressivity at all).
> - The ability to approximate positional encodings, for example for use with hybrid architectures ([1] discusses neither positional encodings nor hybrid architectures/ graph transformers)
>
> Empirically, we believe that the following can be considered our main insights when compared to [1]:
> - Unitary architectures provide excellent MPNN backbones in hybrid architectures and beat non-unitary architectures at capturing topological information (Tables 3, 13, 14, 20)
> - Perfect unitarity is not always required, and sometimes not even beneficial. Reducing the order of the taylor approximation for the exponential map can reduce memory requirements and runtime while simultaneously providing similar or even better performance, especially on homophilous node classification tasks (Tables 4, 5, 6).
>
> > complex numbers in eq. (4)
>
> In line with prior work, we experimented with several ways to project the complex-valued output of equation (5) back onto the real line (including using a complex number's modulus or its real or imaginary part). We ultimately found using the real component of each complex number to work best, but differences were generally small.
>
> > It is not specified how a complete neural network is defined
>
> Following the projection step above, the convolution layer in equation (5) can be treated like any other graph convolution layer and included in a larger GNN. For example, in line with established GNN practice, equation (5) can be followed by a small MLP, a skip-connection, and a non-linearity. The output of this can then be used as input for the next convolution/ message-passing layer. We will clarify this and the projection step in Section 4.1.

---

> > ### Author Rebuttal · Reviewer_41ew · 2026-04-02
> >
> > I thank the authors for their elaborate rebuttal. However, some of my concerns remain open, as I detail below.
> >
> > ## Inclusion of edge features - theory
> >
> > I thank the authors for the clarification about oversmoothing and the connection with the CoED framework, which I believe strengthens the motivation behind their design. About expressivity, there exist frameworks to extend WL-expressivity to graphs with edge features (e.g., [1]). Although a comprehensive framework that discusses the expressivity of directed graphs with edge features might be out of scope, the authors should acknowledge [1] and discuss its relation with their work.
> >
> > [1] Beddar-Wiesing, Silvia, et al. "Weisfeiler–Lehman goes dynamic: An analysis of the expressive power of graph neural networks for attributed and dynamic graphs." _Neural Networks_ 173 (2024): 106213.
> >
> > ## Inclusion of edge features - experiments
> >
> > I thank the authors for pointing to the experiments in Tables 15 and 20. However, these results require better explanations.
> > - GINE and GatedGCN are not accompanied by a citation, the datasets are not described and the experimental settings are not reported.
> > - GINE and GatedGCN are the only edge-aware methods compared. I believe further comparisons would be beneficial, e.g., with [1,2].
> > - These graphs are undirected, so these experiments do not evaluate how DUNE jointly processes directed edges and edge features.
> >
> > I also thank the authors for providing additional experiments. I believe the comparisons between DUNE (e), DUNE (d), DUNE (e,d) help clarify the impact of the different components, and extending this to all datasets with edge features and edge directionality will improve evaluation. However, I have some concerns:
> > - What are the edge features in DSP and LUT?
> > - From the experiments in Table 3, it seems that these datasets do not have edge features. If they do, then I believe the experiments in Table 3 should also be adapted to account for them, edge-aware baselines should be added and this should be clarified in the text.
> >
> > [1] Corso, Gabriele, et al. "Principal neighbourhood aggregation for graph nets." _Advances in neural information processing systems_ 33 (2020): 13260-13271.
> >
> > [2] Simonovsky, Martin, and Nikos Komodakis. "Dynamic edge-conditioned filters in convolutional neural networks on graphs." _Proceedings of the IEEE conference on computer vision and pattern recognition_. 2017.
> >
> > ## Other theoretical results
> >
> > I thank the authors for clarifying their theoretical contributions compared to prior work. This concern is addressed.
> >
> > ## Complex numbers manipulation
> >
> > I thank the authors for clarifying this implementation aspect. However, I still have some concerns.
> > - Is considering only the real component of each complex number equivalent to setting $\alpha=0$ in eq. (4)? If so, then I believe the presentation framework should be modified and not mention complex numbers at all. Otherwise, it would be beneficial to clarify the impact of the imaginary part of these complex number on the overall computation.
> > - How is the expressivity affected by the treatment of complex numbers? It seems that if the imaginary part is discarded, then the DUNE only has access to the unique directed edges of the graph, so its expressivity might reduce.
> > - In any case, the treatment of complex numbers should be made explicit in the text.
> >
> > ## How is the NN implemented
> >
> > I thank the authors for clarifying this implementation aspect. I believe this should be explicitly mentioned in the paper.

---

> > > ### Author Response · Authors · 2026-04-03
> > >
> > > We thank the reviewer for their engagement with our work. We address all remaining concerns below:
> > >
> > > > edge features - theory
> > >
> > > We thank the reviewer for this suggestion. We agree and will cite Beddar-Wiesing et al. in the revision. It is the natural reference point for DUNE’s edge-feature mechanism, since it gives a WL-style framework for graphs with edge features. It does not, however, address the directed setting, whereas our current expressivity result does. We will clarify this relation.
> > >
> > > > edge features - experiments
> > >
> > > We thank the reviewer for pointing out these oversights. We will add citations for GINE [1], GatedGCN [2], and additional edge-aware baselines, and expand the discussion around Tables 15 and 20 to state the datasets, protocol, and evaluation setting more clearly (we follow [3] for comparability). The reviewer is also correct that Tables 15 and 20 only test edge features on undirected graphs, not the joint use of directionality and edge features; we will make this explicit.
> > >
> > > The purpose of those tables was to isolate whether DUNE’s edge-feature mechanism remains competitive with standard edge-aware MPNNs and improves over UniGCN, which does not use edge features. The new directionality/edge-feature ablation we provide therefore addresses a different and complementary question.
> > >
> > > We also agree that broader comparisons would strengthen this part of the paper. We will include PNA and ECC as additional baselines in the final manuscript.
> > >
> > > [1] Hu, Weihua, et al. "Strategies for Pre-training Graph Neural Networks." International Conference on Learning Representations.
> > >
> > > [2] Bresson, Xavier, and Thomas Laurent. "Residual gated graph convnets." arXiv preprint arXiv:1711.07553 (2017).
> > >
> > > [3] Kiani, Bobak, Lukas Fesser, and Melanie Weber. "Unitary convolutions for learning on graphs and groups." Advances in Neural Information Processing Systems 37 (2024)
> > >
> > > > edge features in DSP and LUT
> > >
> > > DSP and LUT do contain edge features: in the HLS benchmark, each edge has a categorical edge-type ID and a binary back-edge indicator [4]. Likewise, Gain, BW, and PM also contain edge information: in the circuit benchmark, edges correspond to component/connectivity types, with 24 categorical edge classes [5]. We will make this explicit in Section 5.2.
> > >
> > > All hybrid backbones in Table 3 except UniGCN were run with edge features, but the current labels obscure this fact. We will therefore replace names such as GIN/GINE or GatedGCN/GatedGCNE with explicit descriptors such as “(e),” “(d),” and “(e,d).” Table 3 was intended to study the role of positional encodings in hybrid DUNE models, not to disentangle positional encodings, directionality, and edge features. We will clarify this in the final manuscript.
> > >
> > > To address the reviewer’s concern more directly, we provide additional hybrid-model results on DSP and LUT below. These serve as an explicit ablation complementing Table 3, focusing on the joint contribution of edge features and directionality without positional encodings. We will extend these experiments, together with the MPNN experiments from the original rebuttal, to all five Table 3 datasets.
> > >
> > > | Model | DSP | LUT |
> > > |-|-:|-:|
> > > | SAT-UniGCN | 2.891 | 2.315 |
> > > | SAT-DUNE (e) | 2.803 | 2.254 |
> > > | SAT-DUNE (d) | 2.728 | 2.179 |
> > > | SAT-DUNE (e,d) | 2.612 | 2.021 |
> > > | SAT-GIN | 3.294 | 2.567 |
> > > | SAT-GIN (e) | 3.242 | 2.492 |
> > > | SAT-BiGIN (d) | 2.860 | 2.318 |
> > > | SAT-BiGIN (e,d) | 2.789 | 2.231 |
> > > | SAT-GatedGCN | 3.167 | 2.508 |
> > > | SAT-GatedGCN (e) | 2.985 | 2.417 |
> > > | SAT-BiGatedGCN (d) | 2.812 | 2.273 |
> > > | SAT-BiGatedGCN (e,d) | 2.741 | 2.232 |
> > >
> > > [4] Wu, Nan, et al. "High-level synthesis performance prediction using gnns: Benchmarking, modeling, and advancing." Proceedings of the 59th ACM/IEEE Design Automation Conference. 2022.
> > >
> > > [5] Dong, Zehao, et al. "CktGNN: Circuit Graph Neural Network for Electronic Design Automation." The Eleventh International Conference on Learning Representations.
> > >
> > > > Complex numbers manipulation
> > >
> > > We thank the reviewer for raising this point. Taking the real part after propagation is not equivalent to setting $\alpha=0$ in Eq. (4): the complex propagator $\exp(\Pi(A)t)$ is computed first, so both terms in $\Pi(A)$ affect the representation, and the real part is only a final projection back to $\mathbb{R}$. We will make this explicit in Section 4.1.
> > >
> > > We also agree that the expressivity claim should be stated more carefully. Our current theorem concerns the DUNE update itself, not a specific real-valued readout. We will separate the operator-level argument from the practical real-valued projection and state this limitation explicitly.
> > >
> > > > How is the NN implemented
> > >
> > > We agree and will make the pipeline explicit: DUNE applies the unitary propagation in Eq. (5), projects the complex output to real features, and then proceeds with standard GNN components such as MLPs, skip connections, and nonlinearities.
> > >
> > > Overall, we thank the reviewer for these suggestions and believe they will make the paper substantially clearer.

---

### Official Review · Reviewer_ZQze · 2026-03-12

**Soundness:** 3
**Presentation:** 4
**Significance:** 3
**Originality:** 3
**Overall Recommendation:** 4
**Confidence:** 3

**Summary:**

This paper introduces Dune, a directed unitary GNN with edge features that extends unitary convolutions to directed graphs through Hermitian projection of the adjacency matrix into the unitary Lie algebra followed by matrix exponential. It preserves unitarity to ensure bounded gradients at arbitrary depth and provably prevents exponential oversmoothing, while naturally incorporating directionality and edge features. The wavelike propagation provides rich geometric inductive bias that reduces reliance on traditional positional encodings in hybrid graph transformers. Experiments across 12 directed benchmarks demonstrate state-of-the-art performance with gains up to 18% over baselines and continued improvement beyond 100 layers, addressing the early saturation typical of prior directed GNNs. The work is supported by solid theoretical analysis and ablations.

**Compliance With Llm Reviewing Policy:**

Affirmed.

**Final Justification:**

The author completely addressed my concerns.I wiil keep my score.

**Key Questions For Authors:**

Why use the fixed α = β = 1/2 in the Hermitian projection? Have you tested other values or different ways to map to unitary operators, and how do they impact results? What is the real computational cost (time/memory) of the matrix exponential in deep models (>50–100 layers) compared to standard GNN layers, especially on large graphs?

**Limitations:**

While Dune offers strong theoretical guarantees and excellent performance on directed graphs, it has limitations: the fixed Hermitian projection parameters lack sensitivity analysis, matrix exponential incurs higher computational cost without detailed efficiency comparisons for deep or large-scale models, experiments are limited to moderate-sized standard benchmarks rather than massive real-world graphs, and hybrid transformer results remain preliminary with insufficient comparison to recent directed positional encodings.

**Strengths And Weaknesses:**

Strengths

The paper presents Dune, a strong directed GNN that extends unitary convolutions to handle directionality and edge features well. It has solid theory showing bounded gradients at any depth and no exponential oversmoothing. The wave-like propagation gives useful geometric information, reducing the need for traditional positional encodings in hybrid models. Experiments show new best results on 12 directed benchmarks (up to +18% better) and the model works well even with over 100 layers.

Weaknesses

The Hermitian projection uses fixed parameters with no clear study on how sensitive it is to changes. Experiments use standard datasets but lack very large real-world examples. The computational cost of matrix exponential is not deeply compared, especially for very deep models. Hybrid transformer results are good but still quite preliminary.

---

> ### Author Rebuttal · Authors · 2026-03-30
>
> We would like to thank the reviewer for their positive and constructive feedback.
>
> > While Dune offers strong theoretical guarantees and excellent performance on directed graphs, it has limitations: the fixed Hermitian projection parameters lack sensitivity analysis…
>
> We thank the reviewer for bringing this up. Reviewer brpQ had a similar question about making alpha and beta in the Hermitian projection learnable, so we present the same additional results with various fixed values of alpha and beta and with alpha and beta learnable here:
>
> | Aggregation                  |                                 Citeseer |                               OGBN-Arxiv |                             Roman Empire |
> | ---------------------------- | ---------------------------------------: | ---------------------------------------: | ---------------------------------------: |
> | $\alpha = 1/2,\ \beta = 1/2$ |                                    94.39 |                                    72.16 |                                    92.58 |
> | $\alpha = 1,\ \beta = 1/2$   |                                    94.18 |                                    71.32 |                                    87.69 |
> | $\alpha = 1/2,\ \beta = 1$   |                                    93.15 |                                    69.48 |                                    85.55 |
> | $\alpha = 1,\ \beta = 0$     |                                    94.43 |                                    69.93 |                                    84.60 |
> | $\alpha = 0,\ \beta = 1$     |                                    81.57 |                                    65.09 |                                    79.73 |
> | trainable                    | 94.24 ($\alpha = 1.011,\ \beta = 1.004$) | 72.90 ($\alpha = 0.934,\ \beta = 0.543$) | 93.82 ($\alpha = 1.145,\ \beta = 0.784$) |
>
> We present the mean over ten runs. In the last row, alpha and beta are learnable, initialized as 0.5, 0.5, and shared across layers. We also experimented with layer-specific learnable values for alpha and beta but found small deviations between layers and no impact on overall performance compared to the shared alpha and beta baseline.
>
> Overall, making alpha and beta learnable can provide performance boosts on OGBN-Arxiv and Roman Empire. It is also noteworthy that a non-zero alpha seems to be much more important than a non-zero beta.
>
> > matrix exponential incurs higher computational cost without detailed efficiency comparisons for deep or large-scale models, …
>
> Please see Tables 5 and 6 in the appendix, which report memory consumption and runtime as we increase the order of the Taylor approximation for the matrix exponential (including on the large OGBN-Arxiv graph with more than a million nodes). Both memory consumption and runtime scale linearly in the number of layers.
>
> > experiments are limited to moderate-sized standard benchmarks rather than massive real-world graphs, ...
>
> We politely disagree with this statement. SNAP, for example, has almost 14 million edges and scaling beyond this makes comparisons with other existing approaches difficult as many of them (e.g. DiGCN, MagNet, and Dir-GAT) run into memory constraints.
>
> > Hybrid transformer results are good but still quite preliminary.
>
> We politely disagree with this, as we report results with 3 different hybrid architectures (GPS, SAT, and Exphormer) on 10 different datasets (Tables 3, 13, and 20). This is significantly more than what previous studies report (e.g. [1, 2]). That being said, we would be happy to run any additional experiments with hybrid architectures or other directed positional encodings that the reviewer might have in mind.
>
> [1] Simon Geisler, Yujia Li, Daniel J Mankowitz, Ali Taylan Cemgil, Stephan Günnemann, and Cosmin Paduraru. Transformers meet directed graphs. In International conference on machine learning, pp.11144–11172. PMLR, 2023.
>
> [2] Huang, Yinan, Haoyu Peter Wang, and Pan Li. "What Are Good Positional Encodings for Directed Graphs?." The Thirteenth International Conference on Learning Representations.

---

> > ### Author Rebuttal · Reviewer_ZQze · 2026-04-06
> >
> > The author completely addressed my concerns.

---

### Official Review · Reviewer_brpQ · 2026-03-13

**Soundness:** 4
**Presentation:** 3
**Significance:** 3
**Originality:** 4
**Overall Recommendation:** 5
**Confidence:** 3

**Summary:**

This paper introduces Dune, a novel Graph Neural Network (GNN) that extends Unitary Graph Convolutions to support directed edges and edge features. While standard directed GNNs often suffer from oversmoothing and gradient instability at depth, and previous unitary GNNs are restricted to undirected graphs, Dune attempts to address both of these limitations.

By using a Hermitian projection to map asymmetric directed adjacency matrices into the Lie algebra of the unitary group, the authors construct a strictly unitary message-passing operator. Theoretically, the paper shows that Dune is as expressive as the Directed Weisfeiler-Lehman (D-WL) test, avoids oversmoothing, and maintains bounded gradients. Empirically, the model achieves good, competitive results across 12 directed graph benchmarks, demonstrates stable training at depths exceeding 100 layers, and shows potential as a positional encoding mechanism for Graph Transformers.

**Compliance With Llm Reviewing Policy:**

Affirmed.

**Final Justification:**

The authors' detailed rebuttal adequately addressed my initial concerns, and I appreciate their thorough effort in addressing the other reviewers' points as well. I am keeping my score at 5 (Accept).

**Key Questions For Authors:**

Here are a few questions and comments for the rebuttal. Please note that there is no need to run extensive new experiments to address these points—I am primarily just looking for a discussion. Also, feel free to be completely direct with me if any of my questions miss the mark:

- **Deviation from unitarity:** Because the exact matrix exponential is intractable and approximated via a truncated Taylor series, the resulting operator is not strictly unitary in practice. Have you measured the actual deviation from unitarity (e.g., $|| U^\dagger U - I ||$) during the forward pass?
- **Learnable projection parameters:** The Hermitian projection currently fixes $\alpha = \beta = 1/2$. Did you experiment with making $\alpha$ and $\beta$ learnable parameters (whether globally, per-layer, or per-dataset)? It seems plausible that different tasks might benefit from dynamically shifting the balance between symmetric and anti-symmetric information flow.
- **Edge feature bottleneck:** Mapping multi-dimensional edge features into a single complex scalar $f_E(E_{ij})$ seems like it could act as an information bottleneck compared to standard message-passing architectures that update full edge embeddings. What impact do you think this could have on tasks that heavily rely on rich edge features?
- **Sensitivity to $K$:** When using Dune as a positional encoding for Graph Transformers, have you analyzed how sensitive the downstream performance is to the degree $K$ of the polynomial expansion?

**Limitations:**

Yes.

**Strengths And Weaknesses:**

###

Overall, I have a positive impression of this paper. The authors tackle a relevant problem: extending unitary graph convolutions to directed graphs with edge features. A notable contribution is the theoretical foundation—using a Hermitian projection to map the asymmetric adjacency matrix into the Lie algebra of the unitary group, which aims to prevent oversmoothing and vanishing or exploding gradients at depth. However, I must explicitly acknowledge that I do not have a high confidence in my assessment of the theoretical proofs, particularly the Lie group theory and continuous dynamics involved.

**Strengths**

- **Theoretical Guarantees for Deep GNNs:** The paper provides theoretical claims that Dune avoids the exponential oversmoothing that often affects existing directed GNNs and maintains bounded gradients. This is empirically validated by training the model stably with over 100 layers.
- **Extension to Directed Graphs:** Adapting unitary convolutions to directed graphs via the Hermitian projection is an elegant solution to a known limitation of prior works like UniConv.
- **Connection to Positional Encodings:** The paper demonstrates that the unitary operator can implicitly capture bidirectional walk profiles. This serves as a positional encoding for hybrid Graph Transformers that reduces the reliance on traditional Laplacian or random-walk encodings.
- **Empirical Performance:** The model achieves good, competitive results across an evaluation suite of 12 directed graph benchmarks, showing its practical utility.

**Weaknesses**

- **Theory vs. Practice Gap:** The theoretical guarantees of unitarity rely on an exact matrix exponential, but the practical implementation uses a truncated Taylor approximation. It is not clear how this truncation error impacts the theoretical guarantees, especially when errors might accumulate over many layers (though results suggest it is fairly ok).
- **Fixed Constants:** The projection uses fixed constants $\alpha = \beta = 1/2$. Because these control the balance between symmetric and directional information flow, fixing them seems arbitrary. It would be beneficial to see if making them learnable parameters yields better dataset-specific adaptations.
- **Edge Feature Compression:** To incorporate edge features, the approach maps multi-dimensional edge features into a complex scalar before convolution. This might act as an expressivity bottleneck compared to standard MPNNs that maintain and update high-dimensional edge representations throughout the network.
- **Scalability on Large Graphs:** While evaluating polynomials of adjacency matrices avoids full $O(N^3)$ eigendecomposition, it can still become memory-intensive on large graphs. Authors, however, identify this as a shared scalability limitation of existing directed solutions.

In conclusion, I lean towards acceptance. The core idea is interesting, the theoretical properties are desirable, and the empirical results demonstrate good performance. I am looking forward to seeing the discussions with other reviewers during the rebuttal phase. Please feel free to use straightforward language in your replies to me with zero problems.

---

> ### Author Rebuttal · Authors · 2026-03-30
>
> We would like to thank the reviewer for their positive and constructive feedback.
>
> > Deviation from unitarity: Because the exact matrix exponential is intractable and approximated via a truncated Taylor series, the resulting operator is not strictly unitary in practice. Have you measured the actual deviation from unitarity (e.g., ) during the forward pass?
>
> One can prove that the approximation error shrinks exponentially fast in the order of the Taylor series. We verify this empirically on random directed graphs and present the results below:
>
> | T  | mean error (Frobenius) | standard dev. |
> | -- | ---------------------: | -----------------: |
> | 1  |           2.051141e+00 |       1.369666e-01 |
> | 2  |           4.856151e-01 |       7.707620e-02 |
> | 3  |           9.864454e-02 |       9.996611e-03 |
> | 4  |           2.771307e-02 |       6.485767e-03 |
> | 5  |           3.681626e-03 |       6.931195e-04 |
> | 6  |           9.042888e-04 |       2.911879e-04 |
> | 7  |           7.969472e-05 |       2.116411e-05 |
> | 8  |           1.715961e-05 |       6.972042e-06 |
> | 9  |           1.131391e-06 |       3.894751e-07 |
> | 10 |           2.140336e-07 |       1.049672e-07 |
>
> > Learnable projection parameters: The Hermitian projection currently fixes. Did you experiment with making and learnable parameters (whether globally, per-layer, or per-dataset)? It seems plausible that different tasks might benefit from dynamically shifting the balance between symmetric and anti-symmetric information flow.
>
> We thank the reviewer for raising this excellent point. We had originally not experimented with this, but are happy to provide some additional results with various fixed values of alpha and beta and with alpha and beta learnable below:
>
> | Aggregation                  |                                 Citeseer |                               OGBN-Arxiv |                             Roman Empire |
> | ---------------------------- | ---------------------------------------: | ---------------------------------------: | ---------------------------------------: |
> | $\alpha = 1/2,\ \beta = 1/2$ |                                    94.39 |                                    72.16 |                                    92.58 |
> | $\alpha = 1,\ \beta = 1/2$   |                                    94.18 |                                    71.32 |                                    87.69 |
> | $\alpha = 1/2,\ \beta = 1$   |                                    93.15 |                                    69.48 |                                    85.55 |
> | $\alpha = 1,\ \beta = 0$     |                                    94.43 |                                    69.93 |                                    84.60 |
> | $\alpha = 0,\ \beta = 1$     |                                    81.57 |                                    65.09 |                                    79.73 |
> | trainable                    | 94.24 ($\alpha = 1.011,\ \beta = 1.004$) | 72.90 ($\alpha = 0.934,\ \beta = 0.543$) | 93.82 ($\alpha = 1.145,\ \beta = 0.784$) |
>
> We present the mean over ten runs. In the last row, alpha and beta are learnable, initialized as 0.5, 0.5, and shared across layers. We also experimented with layer-specific learnable values for alpha and beta but found small deviations between layers and no impact on overall performance compared to the shared alpha and beta baseline. Overall, making alpha and beta learnable can provide performance boosts on OGBN-Arxiv and Roman Empire. It is also noteworthy that a non-zero alpha seems to be much more important than a non-zero beta. We will extend these experiments to all datasets and include the results in the final manuscript.
>
> > Edge feature bottleneck: Mapping multi-dimensional edge features into a single complex scalar seems like it could act as an information bottleneck compared to standard message-passing architectures that update full edge embeddings. What impact do you think this could have on tasks that heavily rely on rich edge features?
>
> To the best of our knowledge, the most common MPNNs with edge features (e.g. GINE, GatedGCN) do not actively update edge features, so we did not experiment with this. The reviewer is right that in some settings, this might become a performance bottleneck, which is why we believe that this is a promising direction for future work.
>
> > When using Dune as a positional encoding for Graph Transformers, have you analyzed how sensitive the downstream performance is to the degree of the polynomial expansion?
>
> Let $K$ be the depth of the Dune model used and let $T$ be the order of the polynomial approximation. Then in synthetic experiments (as in Table 14), we empirically observe a performance degradation if $TK$ is smaller than the walk length we wish to capture. On real-world datasets (e.g. Table 3), this effect seems much less pronounced, and performance is generally insensitive to changing $T$ once $TK$ is above ~10 (which is most likely an artifact of the datasets used).

---

> > ### Author Rebuttal · Reviewer_brpQ · 2026-04-04
> >
> > Thank you for the detailed rebuttal. You have adequately addressed my concerns, and I appreciate the thorough effort you are putting into addressing the other reviewers' points as well. I am keeping my score at 5.

---

### Official Review · Reviewer_XR1b · 2026-03-22

**Soundness:** 3
**Presentation:** 2
**Significance:** 2
**Originality:** 2
**Overall Recommendation:** 3
**Confidence:** 3

**Summary:**

The paper presents DUNE, a new directed unitary GNN that incorporate edge features and edge directionality, together with unitary graph convolution. This extension represents a principled construction for handling data that lives on directed networks and graphs, and permits to prevents overshooting phenomenons that appear in naive architectures for directed GNNs. Experimental results on multiple benchmark datasets confirms the benefits of the proposed architecture.

**Compliance With Llm Reviewing Policy:**

Affirmed.

**Final Justification:**

The rebutall discussion has been insightful, but unfortunately not fully convincing about novelty aspects.

**Key Questions For Authors:**

see above

**Limitations:**

yes

**Strengths And Weaknesses:**

Strengths

1. The contribution is generally good, yet it is a relatively simple extension of previous work in terms of methodology. Unitary neural networks have been proposed in different flavours already, not in the exact settings considered in this paper. Yet, the novelty of the proposed contribution with respect to UniGCN in particular, remains relatively limited.

2. The work is generally quite complete, and relatively convincing in the extended experimental results section.  (However, the statement 'consistently outperforms' in the discussion of page 8, seems a bit strong wrt to the actual results shown in Table 2 especially)

3. The presentation is generally quite good, and the discussion of the method and the analysis of the results are generally quite clear.

Weaknesses:

1. Several works have proposed recently different flavours of graph representation learning in generative settings for directed graphs: it would be good to discuss those recent works too, and possibly discuss the differences in the design of the representation framework.

2. Proposition 3 seems to be a simple extension of the results of Dir-GNN. It may be good to either emphasize the main difference in the proof construction, or maybe drop this proposition and simply refer to Dir-GNN.

3. Proposition 5 states that Dune 'does not oversmooth'. What is the actual definition of oversmoothing? as used inn this proposition, and following ones?

4. Proposition 10 may appear a bit out of context

5. The role of transformers in the properties described in Section 4, is not clearly discussed. Same for position encoding, whose design is not presented in details unfortunately, which makes it difficult to appreciate its role, as well as the experimental study that specifically considers the influence of positional encoding in the proposed framework.

Overall, the contribution is generally interesting, even if the methodological novelty does not appear to be dramatic. Also, the positioning of the proposed framework with respect to recent works in generative models in particular, should be discussed, and a more precise description of the role of positional encoding, and transformer architectures, should be provided.

---

> ### Author Rebuttal · Authors · 2026-03-30
>
> We thank the reviewer for their constructive feedback. As we understand it, the comments primarily concern clarification and presentation rather than requests for additional experiments or major technical issues. We are happy to incorporate all of these improvements in the final manuscript. If the reviewer has any further substantive concerns that influenced their overall assessment, we would greatly appreciate clarification and would be glad to address them.
>
> > Several works have proposed recently different flavours of graph representation learning in generative settings for directed graphs: it would be good to discuss those recent works too, and possibly discuss the differences in the design of the representation framework.
>
> Could we ask the reviewer to clarify what they mean by 'graph representation learning in generative settings for directed graphs' and perhaps point us to 1-2 representative publications? We would be happy to discuss and cite recent works that the reviewer believes are relevant to our work.
>
> > Proposition 3 seems to be a simple extension of the results of Dir-GNN. It may be good to either emphasize the main difference in the proof construction, or maybe drop this proposition and simply refer to Dir-GNN.
>
> The main difference as we see it is the Taylor expansion to the exponential map, which Dir-GNN does not have, and the fact that Dir-GNN uses two different weight matrices for incoming and outgoing edges, while Dune uses the same weight matrix for both directions. We therefore thought it better to include a full proof as opposed to simply pointing to Dir-GNN and letting the reader fill in the details.
>
> > Proposition 5 states that Dune 'does not oversmooth'. What is the actual definition of oversmoothing? as used inn this proposition, and following ones?
>
> Please see Definition 5, in particular Equation 11.
>
> > Proposition 10 may appear a bit out of context
>
> > The role of transformers in the properties described in Section 4, is not clearly discussed. Same for position encoding, whose design is not presented in details unfortunately, which makes it difficult to appreciate its role, as well as the experimental study that specifically considers the influence of positional encoding in the proposed framework.
>
> We agree with the reviewer that section 4.3 could benefit from more details on (hybrid) graph transformers and positional encodings, both for improved accessibility and to better appreciate Proposition 10. We will move these details, which are currently in appendices A.3 and A.4, to the main text for the final manuscript.

---

> > ### Author Rebuttal · Reviewer_XR1b · 2026-04-02
> >
> > Thanks for the response, which does unfortunately not manage to fully convince that the proposed contribution meets the novelty threshold of conferences like ICML.
> >
> > Some literature on generative models for directed graphs:
> >
> > Zhang et.al  D-VAE: A Variational Autoencoder for Directed Acyclic Graphs. NeurIPS 2019
> > Li et al. LayerDAG: A Layerwise Autoregressive Diffusion Model for Directed Acyclic Graph Generation. ICLR 2025.
> > Law et al. Directed Graph Generation with Heat Kernels. TMLR 2025.
> >
> > This is certainly not an exhaustive list, and not all references are equally relevant; yet those might be helpful to position the proposed contribution more broadly.

---

> > > ### Author Response · Authors · 2026-04-02
> > >
> > > Thank you for the follow-up and for pointing us to these references. We agree that the paper would benefit from broader positioning with respect to recent work on directed graphs, and we will add a short discussion of these generative models in the final version. At the same time, we would like to clarify that our contribution is in **directed representation learning** rather than graph generation. We also agree that our contribution is not a radical departure from prior unitary GNN work. Rather, its novelty is in extending that line to a new setting—directed representation learning with edge features—and in showing that this extension yields both new theory and strong empirical benefits.
> > >
> > > In particular, relative to UniGCN and prior unitary graph models, we believe the main novelties are:
> > >
> > > - a **unitary message-passing operator for directed graphs with edge features**, rather than the undirected setting considered in UniGCN;
> > > - a **theoretical analysis specific to the directed setting**, including expressivity, oversmoothing behavior, and stable gradient propagation;
> > > - the use of Dune not only as a stand-alone MPNN, but also as a **backbone in hybrid graph-transformer architectures**;
> > > - a connection between the unitary operator and **positional-encoding-like geometric information** on directed graphs, supported by both theory and experiments.
> > >
> > > We will revise the manuscript to make this positioning more explicit and to better distinguish our contribution from both earlier unitary GNNs and the broader directed-graph literature. We hope this clearer framing also addresses the novelty concern underlying your current assessment.

---

### Decision · Program_Chairs · 2026-04-30

**Decision:**

Accept (regular)

**Comment:**

The paper introduces DUNE, a directed unitary GNN that extends unitary graph convolutions to directed graphs with edge features via a Hermitian projection of the asymmetric adjacency into the unitary-group Lie algebra. It provably avoids exponential oversmoothing and maintains bounded gradients at arbitrary depth, and the wavelike propagation supplies geometric positional information that reduces reliance on random-walk or Laplacian-based encodings in hybrid graph transformers.

The paper provides a principled and theoretically grounded operator for directed graphs, solid empirical results, and a useful connection to positional encodings in graph transformers. Three of four reviewers recommend acceptance. The rebuttal, which included additional ablations and scalability experiments, moved most reviewers to "fully resolved." I recommend acceptance.

For the camera-ready: discussion of the edge-feature bottleneck for tasks with rich edge embeddings, clarification that only the real component of the complex output is used and its implications for expressivity, incorporation of additional heterophilic-baseline comparisons from the rebuttal, and expanded scalability discussion.